

# Association of heat shock protein polymorphisms with patient susceptibility to coronary artery disease comorbid depression and anxiety in a Chinese population

Haidong Wang[1,*], Yudong Ba[2,*], Wenxiu Han[3], Haixia Zhang[3], Laiqing Zhu[3] and Pei Jiang[3]

[1] Department of Pharmacy, The Affiliated Lianyungang Hospital of Xuzhou Medical University/The First People's Hospital of Lianyungang, Lianyungang, China
[2] Department of Pharmacy, Dongying People's Hospital, Dongying, China
[3] Jining First People's Hospital, Jining Medical University, Jining, China
[*] These authors contributed equally to this work.

Corresponding authors
Laiqing Zhu, jnlaiqingzhu@yeah.net
Pei Jiang, jiangpeicsu@sina.com

## ABSTRACT

**Background.** Coronary artery disease (CAD) is one of the severe diseases that threaten human health worldwide. In addition, the associated rate of comorbidity with depression and anxiety is extremely high. Heat shock proteins (HSPs) are a group of proteins that possesses cardiovascular and psychological protection properties. The objective of this study is to determine the association of the two most widely studied HSPs, namely, HSP70 and HSP90, with CAD comorbid depression and anxiety in a Chinese population.

**Methods.** A case-control study involving 271 CAD patients and 113 healthy individuals was conducted. The 271 CAD patients include individuals with (123) and without depression (148) and individuals with (57) and without anxiety (214). Ten single nucleotide polymorphisms (SNPs) for HSP70 and seven SNPs for HSP90 were selected and genotyped.

**Results.** Results revealed that the HSP70 rs10892958 C allele and HSP70 rs2236658 T allele were associated with a decreased risk of CAD ($P < 0.05$), whereas the G allele of the rs11218941 polymorphism was associated with an increased risk of CAD. The haplotype analysis results indicated that the haplotype TGGGC of the HSPA8 gene (coded the HSP70 family, rs4936770/rs4802/rs10892958/rs11218941/rs2236658) significantly increased the risk of CAD ($P = 0.008$). Among the patients with CAD, the carriers of the CC genotype for the HSP90 rs1042665 showed higher risks of anxiety than the carriers of another genotypes. However, no significant relationships were found among the CAD with depression and CAD without depression groups for the selected SNPs. These findings suggested that the genetic polymorphisms in the HSP gene, especially the HSPA8 of HSP70, contribute to CAD susceptibility and rs1042665 genetic polymorphisms might have an effect on the anxiety incidence among CAD patients.

## INTRODUCTION

Cardiovascular diseases (CVDs) are the number one cause of mortality in the world. These diseases account for one-third of the total deaths and approximately 17.9 million lives each year. Coronary artery disease (CAD) is an atherosclerotic CVD manifested by stable or unstable angina, myocardial infarction (MI), or sudden cardiac death. CAD is the leading cause of death in developed and developing countries (*Malakar et al., 2019*; *Roth et al., 2017*). More than 900,000 individuals in the United States suffered from MI or died of CAD in 2017 (*Benjamin et al., 2018*). CAD-driven death ranked first in six provinces in China. Patients with CAD experience high levels of mental disorder, which might further influence the course of illness and the recovery of the physical and psychological states. Anxiety and depression are highly prevalent among CAD patients and cause adverse outcomes, which affect over 20% and 30% of patients with CAD, respectively.(*Gu et al., 2016*; *Rudisch & Nemeroff, 2003*) *Wang et al. (2013)* reported that anxiety is independently associated with the severity of coronary atherosclerosis and predicted the worse outcome in CAD patients with Han Chinese ethnicity. Depressive disorders also increase the mortality in CAD patients and the risk of developing the disease. Therefore, increasing attentions have been devoted to the importance of comorbid anxiety and depression among CAD patients.

The relationship between psychiatric disorders and CAD is multidimensional. Several potential mechanisms were posited. *Stapelberg et al. (2011)* conducted a thorough review and proposed six domains of causal mechanism for the potentially linked depression and cardiac disease, namely, behavioral mechanisms, genetic mechanisms, inflammatory mechanisms, endothelial dysfunction and platelet activation, polyunsaturated omega-3 free fatty acid deficiency and hypothalamic–pituitary–adrenal axis, and autonomic mechanisms. Numerous studies have focused on the genetic mechanisms of the relationship between psychiatric diseases and CAD (*Amare et al., 2019*; *Liu et al., 2014*; *Meyer et al., 2020*).

Heat shock proteins (HSPs) are a group of highly conserved stress proteins whose expression is stimulated by heat shock, inflammation, infection, ischemia hypoxia, physiological stress, and carcinogens. Many HSPs have a molecular chaperone activity involved in various aspects of protein biogenesis, including folding, oligomeric assembling, transporting to a particular subcellular compartment, or controlled switching among active/inactive conformations. In addition, HSPs are governed tightly by cellular regulatory mechanism and get expressed under varying metabolic stress, including CAD, anxiety, depression and other psychiatric diseases, protecting cells and tissue from misfolding of denatured proteins by regulating both transcription and translation (*Dhama et al., 2019*). HSPs are classified into six families in accordance with their molecular weight, which ranges from 17 kDa to more than 100 kDa. These families include small HSP, HSP40,

HSP60, HSP70, HSP90, and HSP100 families. Out of these six, HSP70 and HSP90 are the most well-known families.

Various studies have reported the cardiovascular and psychological regulatory effects of HSP70 and HSP90. The phosphorylated HSP70 provides an on/off switch for the regulation of Ca signaling, which leads to therapeutic benefits in human diseases, such as atherosclerosis, cardiomyopathy, congestive heart failure and ischemia (*Lakshmikuttyamma, Selvakumar & Sharma, 2006*). *Song, Zhong & Wang (2019)* highlighted that HSP70 is a promising therapeutic target for myocardial ischemia–reperfusion injury. Fibronectin levels and collagen production would be decreased by inhibiting the extracellular HSP90, implicating for the fibrosis-related pathology of chronic heart conditions (*García et al., 2016*). Moreover, extracellular HSP70 and HSP90 are considered as potential diagnostic markers of chronic heart disease (*Edkins et al., 2018*).

Additionally, the abnormal functioning of HSP70 is hypothesized as an underlying target for altered stress response in major depression (*Cheng et al., 2018*). The genetic variants of the HSP70 family proteins may affect the action of antidepressants (*Pae et al., 2007*). HSP90, a molecular chaperone of FK-506 binding protein 5 (FKBP5), coordinates with FKBP5 to regulate glucocorticoid receptor (GR) activity; both are factors of several affective disorders, including anxiety, depression, and post-traumatic stress disorder (PTSD) (*Criado-Marrero et al., 2018*).

Numerous studies focused on the relationship between HSP70 gene polymorphism and CAD. It stated that several HSP70 SNPs are associated with the risk of developing CAD. Several works reported that rs1061581 (+1267), which is the most studied polymorphism, is associated with CAD. This inference is supported by studies from Iran, Tunisia, Italy, and China (*Giacconi et al., 2006*; *Hrira et al., 2012*; *Mardan-Nik et al., 2016*; *Wei et al., 2014*). *He et al. (2010)* suggested that the genetic variants of HSP70 gene, especially the promoter single nucleotide polymorphism (SNP) rs2236659, contribute to coronary heart disease (CHD) susceptibility by affecting the HSP protein expression levels.

HPS70 and HSP90 show neuroprotective effects in many studies. HSP70 could suppress the progression of Alzheimer's disease with in vitro and in vivo experiments and presented protective effects in neurodegenerative shocks (*Campanella & Pace, 2018*; *Pratt et al., 2015*). *Shimizu et al. (1996)* found abnormal HSP70 mRNA with a 162-base deletion in patients with major depression, whereas another group reported that the 162-base deletion in the 5′-flanking region of the HSP70 gene mRNA is not associated with major depression (*Takimoto et al., 2003*). The result of a study from South Korea reported that the genetic variants within the gene coding for the HSP70 family proteins may affect the action and therapeutic efficacy of antidepressants (*Pae et al., 2007*). *Vlachos et al. (2014)* found the correlation among depression, anxiety, and polymorphonuclear cell resilience in ulcerative colitis, which was mediated by HSP70. Moreover, studies from Taiwan suggested that HSP70 polymorphisms might play protective role against Alzheimer's disease but are unlikely to play a major role in mitigating the risk of developing Parkinson's disease (*Chen et al., 2008*; *Hsu et al., 2008*). A previous review (*Criado-Marrero et al., 2018*) stated that HSP90 served as complex regulators of various psychiatric diseases, such as depression, PTSD, and anxiety.

GR is a crucial part of the stress hormone axis and dysregulation for GR is consistently proven its importance for stress-related diseases, such as depression and anxiety (*Holsboer, 2000*). It has been demonstrated that Hsp90 is required for GR maturation and the related regulation function of HSP90 is primarily accomplished through interactions with its co-chaperones, FKBP5 (FKBP51 and FKBP52). A large number of studies focused their attention on the association between FKBP5 and depression or antidepressant response and Fries's group provided a summary (*Fries, Gassen & Rein, 2017*). Another study overviewed the association of the FKBP5 SNPs with mental health disorders and proposed four FKBP5 SNPs (rs9470080, rs1360780, rs9296158, rs3800373) are significantly associated with at least five diseases, including anxiety and depression (*Criado-Marrero et al., 2018*). One of our recent study also reported the significant association between FKBP5 gene variations and the risk of CAD comorbid depression in a north Chinese population (*Wang et al., 2020*). Given the close molecular chaperone relation between FKBP5 and HSP90, the assumption that the HSP90 gene polymorphisms may affect the susceptibility to CAD comorbid depression/anxiety is reasonable.

The objective of the present study is to address the lack of study on the association of HSP polymorphisms with patient susceptibility to CAD comorbid depression and anxiety. After searching information for related studies (*Lei et al., 2017*; *Sun et al., 2018*; *Zou et al., 2016*), we selected 17 SNPs from HSP70 (rs2763979 (C>T), rs2075800(C>T), rs1380154(C>T), rs391957(C>T), rs17840761(A>G), rs11218941(A>G), rs10892958(G>C), rs2236658(C>T), rs4802(A>G), rs4936770(T>C)) and HSP90 (rs1042665(C>T), rs10873531(A>G), rs8005905(A>T), rs1165678(A>G), rs1165681(C>T), rs1882019 (G>A), and rs3794241(C>T)), which were extensively studied in different diseases and some have been selected as tagSNPs for gene-disease associations, to identify the importance of genetic polymorphisms in the susceptibility to CAD comorbid depression and anxiety .

## MATERIALS & METHODS

### Subjects

The participants in this study were recruited from the First Peoples' Hospital of Jining between May 2016 and July 2019. A total of 271 CAD patients and 113 healthy individuals (controls) were involved in the study. All participants were unrelated North Chinese Han individuals.

The software of Power and Sample Size Calculation (Version 3.1.2, Department of Biostatistics, Vanderbilt University, Nashville, TN, United States) was employed to calculate the sample size. It was calculated that the minimum sample size for experimental subjects and control subjects was 138 and 111, respectively. The calculation was made according to the following parameters: an independent case–control study; a type I error ($\alpha$) of 0.05; a statistical power of 0.8; the probability of exposure in cases (p0) of 0.15; the probability of exposure in the control (p1) of 0.3; and the control to case ratio (m) of 0.8. Therefore, the total of 271 CAD patients and 113 healthy controls basically meet the study requirements.

This study was approved by the ethics committee of the First Peoples' Hospital of Jining (approval number: JY2016062) and adhered to the principles set by the Declaration of Helsinki. Written informed consent was provided by every participant prior to the study.

The CAD cases were diagnosed by experienced cardiologists in accordance with the following diagnostic criteria: (1) at least one of the major segments of the coronary arteries has 50% or higher stenosis based on coronary angiography, (2) history of percutaneous coronary intervention, and (3) history of coronary artery bypass graft surgery. Patients were excluded if they have other serious diseases, such as congenital heart disease, severe autoimmune disease, renal failure, immune system disorders, and malignancies.

The symptoms of depression and anxiety of CAD patients were assessed by two experienced psychiatrists in accordance to the 5th Edition of the Diagnostic and Statistical Manual of Mental Disorders. The Patient Health Questionnaire-9 (PHQ-9) and the seven-item Generalized Anxiety Disorder 7 (GAD-7) were used to assess such symptoms, respectively. A score of 5 or higher in the PHQ-9 scale was selected as the cutoff score for identifying the symptoms of depression, whereas a threshold of >5 points in the GAD-7 scale was set to indicate the possibility of anxiety.

The controls who matched the sex and age of the patients were selected from the medical clinical examination program, which included clinical physical examination, electrocardiogram analysis, radiographic chest examination, and medical history evaluation. After screening, 113 healthy volunteers who met the criteria were included in this study.

## SNP selection and Genotyping

Candidate SNPs in this study were selected as follows: (1) SNPs from public databases and literatures; (2) SNPs that previously were studied in gene-disease association outcomes (for instance, noise-induced hearing loss, cancer risk, bipolar disorder, Alzheimer's disease and diabetes), and (3) SNPs with a minor allele frequency not less than 5%. Finally, 17 SNPs from HSP70 (rs2763979, rs2075800, rs1380154, rs391957, rs17840761, rs11218941, rs10892958, rs2236658, rs4802, rs4936770) and HSP90 (rs1042665, rs10873531, rs8005905, rs1165678, rs1165681, rs1882019, and rs3794241) were selected and genotyped.

First, 4 ml of peripheral blood were collected from each subject, and the genomic DNA was extracted using a TIANamp Blood DNA Kit (TIANGEN, China) in accordance with the manufacturer's instructions. NanoDrop-1000 (NanoDrop, United States of America [USA]) was used to detect the concentration and purity of the DNA samples for the subsequent experiments. Second, all DNA samples were genotyped through polymerase chain reaction (PCR)–ligase detection reaction. Table 1 presents the primers for amplifying the PCR of the target SNPs for each participant. After treating with shrimp alkaline phosphatase, the samples were extended and purified using iPLEX extension reagents (Agena Bioscience, USA) and Nanodispenser RS1000. Finally, the primer extension products were detected through matrix-assisted laser desorption/ionization time-of-flight mass spectrometry, and the genotyping data were automatically analyzed by the SpectroTyper. At least 10% of the samples were randomly selected and retested to verify the reliability of the MassARRAY results.

**Table 1  Primers of HSP genes used in the PCR.**

| SNP | Ancestor allele | Primer sequence | Product size |
|---|---|---|---|
| rs2763979 | C | F: 5′- ACGTTGGATGTCTTACTCGGGACTGTGAGG-3′<br>R: 5′- ACGTTGGATGCACCTCCTTCCTACTTTCTC-3′ | 99 |
| rs2075800 | C | F: 5′- ACGTTGGATGAGGTCAATCAACTGGCAGAG-3′<br>R: 5′- ACGTTGGATGCTCCTTGGTAGAGTTTTGTG-3′ | 116 |
| rs1380154 | T | F: 5′- ACGTTGGATGCAAACCATGTGTTTACTTAC-3′<br>R: 5′- ACGTTGGATGTTGATATGGGACATTCTGCC-3′ | 102 |
| rs391957 | T | F: 5′- ACGTTGGATGATGGAGGAAGGGAGAACAAG-3′<br>R: 5′- ACGTTGGATGAGTAGGTCCAGCAGGAGTGA -3′ | 102 |
| rs17840761 | G | F: 5′- ACGTTGGATGACAAGTCCCGCCTTCACTC-3′<br>R: 5′- ACGTTGGATGAAGTTTCAGATCCCACAGCC-3′ | 105 |
| rs11218941 | A | F: 5′- ACGTTGGATGCAGCGTTTTCTTTCACCCAG -3′<br>R: 5′- ACGTTGGATGATGTACCCCCATACTGGAAG-3′ | 112 |
| rs10892958 | C | F: 5′- ACGTTGGATGGAGGTGATGGGCACTATTAC -3′<br>R: 5′- ACGTTGGATGCCCTCATCCCTTAACAGAAC-3′ | 96 |
| rs2236658 | T | F: 5′- ACGTTGGATGGGGTAACTGAGGACTCCCGC -3′<br>R: 5′- ACGTTGGATGGCGTTCTGGAACTTTCAAGC -3′ | 119 |
| rs4802 | A | F: 5′- ACGTTGGATGGGGGTTGCAAACTTTCTCCAG -3′<br>R: 5′- ACGTTGGATGATTCCTTTTTCTCTTCCTC -3′ | 100 |
| rs4936770 | C | F: 5′- ACGTTGGATGGGGTAAACCAAGCTTGAGCTG-3′<br>R: 5′- ACGTTGGATGTGAATTCTGGTGGAAACCGC-3′ | 103 |
| rs1042665 | T | F: 5′- ACGTTGGATGATTTTCCGTGTCTCACCTGC-3′<br>R: 5′- ACGTTGGATGACAACATGGCACTTCAGAGG-3′ | 107 |
| rs10873531 | G | F: 5′- ACGTTGGATGACTCTGTCTCTGGAAACAGC -3′<br>R: 5′- ACGTTGGATGCACCTTGGCTCTGTCTGAAG -3′ | 100 |
| rs8005905 | T | F: 5′- ACGTTGGATGTCCAGAGACAGAGTAGAGTG-3′<br>R: 5′- ACGTTGGATGGGTACCAAGAAAAGGCCCAAG-3′ | 94 |
| rs1165678 | G | F: 5′- ACGTTGGATGCATCTGCAGAAACGTCTACC -3′<br>R: 5′- ACGTTGGATGTGGAGAGAACTGAGACAGAG -3′ | 92 |
| rs1165681 | T | F: 5′- ACGTTGGATGAGGTAAAGCCAAGACAGAAC -3′<br>R: 5′- ACGTTGGATGTTAGAAGTGGTGACTGCCTC -3′ | 108 |
| rs1882019 | G | F: 5′- ACGTTGGATGAAGATTGGGTAAGGGGCAAC -3′<br>R: 5′- ACGTTGGATGCATGAAAGCACAAGCGTACC -3′ | 117 |
| rs3794241 | C | F: 5′- ACGTTGGATGTAACTCTTAGTGATGCCTCC-3′<br>R: 5′- ACGTTGGATGGGTTCCCAATTTACCTTCCCC -3′ | 99 |

## Statistical analysis

SPSS Statistics version 19.0 (SPSS, Inc., Chicago, IL, USA) was used for the statistical analysis. The continuous and categorical variables were presented as mean ± standard deviation and numbers (%), respectively. The chi-square ($\chi 2$) and $t$-test statistics were used to compare the categorical and continuous variables, respectively. The PHQ-9 and GAD-7 scores that fit a nonparametric distribution were tested for significance by using the two-tailed Mann–Whitney U test. The $\chi 2$ test was also adopted to assess the Hardy–Weinberg equilibrium (HWE) for polymorphisms, the genotype distributions, and the allele frequencies. Pairwise linkage disequilibrium (LD) analyses were conducted using SHEsisPlus (http://shesisplus.bio-x.cn/SHEsis.html). The differences in the risks of CAD and psychiatric diseases among different groups were calculated using a binary logistic

**Table 2  Demographic and clinical characteristics of the participants.**

| Variables | CAD (n = 271) | Controls (n = 113) | P-value[a] | CAD+D (n = 123) | CAD-D (n = 148) | P-value[b] | CAD+A (n = 57) | CAD-A (n = 214) | P-value[c] |
|---|---|---|---|---|---|---|---|---|---|
| Age (yrs) | 58.4 ± 10.1 | 54.1 ± 10.3 | 0.126 | 58.9 ± 10.9 | 57.9 ± 9.5 | 0.147 | 58.1 ± 9.4 | 58.4 ± 10.2 | 0.376 |
| Gender (M/F, n) | 128/143 | 62/51 | 0.173 | 53/70 | 75/72 | 0.213 | 25/32 | 103/111 | 0.566 |
| Smoking (n, %) | 90, 33.2 | – | – | 38, 30.9 | 52, 35.1 | 0.461 | 17,29.8 | 73,34.1 | 0.541 |
| Drinking (n, %) | 97, 35.8 | – | – | 37, 30.1 | 60, 40.5 | 0.074 | 18,31.6 | 79,36.9 | 0.455 |
| Hypertension (n, %) | 210, 77.5 | – | – | 66, 53.7 | 144, 97.3 | 0.000* | 33, 57.9 | 177, 82.7 | 0.000* |
| Diabetes mellitus (n, %) | 53, 19.6 | – | – | 23, 18.7 | 30, 20.3 | 0.657 | 9, 15.8 | 44, 20.6 | 0.420 |
| Stroke (n, %) | 52, 19.3 | – | – | 22, 17.9 | 30, 20.3 | 0.539 | 7, 12.3 | 45, 21.0 | 0.136 |
| Insomnia (n, %) | 95, 35.1 | – | – | 58, 47.2 | 37, 25.0 | 0.000* | 29, 50.9 | 66, 30.8 | 0.005* |

Notes.
[a] CAD versus controls
[b] CAD+D versus CAD-D.
[c] CAD+A versus CAD-A.
CAD, coronary heart disease; CAD+D, CHD with depression; CAD-D, CAD without depression; CAD+A, CAD with anxiety; CAD-A, CAD without anxiety.

regression model. Odds ratios and 95% confidence intervals were also calculated when needed. A two-sided *p* value of $P < 0.05$ was considered statistically significant.

# RESULTS

The $\chi 2$ test results demonstrate that the 17 observed genotype frequencies in the CAD and control groups are in accordance with the HWE ($P \geq 0.111$), which means that the participants represent the target population.

No significant differences ($P > 0.05$) are found in terms of age, gender, smoking habit, or drinking habit between the CAD patients and controls or among the different subgroups divided by comorbid depression (D) or anxiety (A).

## Characteristics of the study participants

The demographic and clinical characteristics of the participants in this study are presented in Table 2. No significant differences ($P > 0.05$) were found in term of age, gender or smoking, drinking, no matter between the CAD patients and the health control or between the different subgroups divided by whether comorbid depression (D) or anxiety (A). The clinical characteristics of CAD patients from these subgroups were also presented in Table 2. The patients with comorbid depression or anxiety had a decreased rate of hypertension relative to those without these psychiatric disorders (53.7% vs. 97.3%, and 57.9% vs. 82.7%, respectively; $P < 0.05$). We also found that CAD patients who had depression or anxiety had markedly higher rates of comorbid insomnia than did those without depression or anxiety (47.2% vs. 25.0% and 50.9% vs. 30.8%, respectively; $P < 0.005$).

## Association between gene polymorphisms and CAD risk

The genotypic distribution and allele frequencies of the 17 genetic polymorphisms between the CAD and control groups were compared. The results are summarized in Tables 3 and 4, respectively. No statistically significant difference is observed between the CAD patients and controls for the genotypic and allelic distributions of rs2763979 (C>T),

**Table 3  Genotypic distribution of 17 HSP gene between all CAD patients ($n = 271$) and controls ($n = 113$).**

| SNP | Genotype | Case, (%) | Control, (%) | P value[a] ($\chi^2$) | P value[b] | OR (95% CI) |
|------|----------|-----------|--------------|----------------------|-----------|-------------|
| rs2763979 | CC | 104(38.4) | 37(32.7) | 0.352(2.088) | Referent | 1.00 |
| | CT | 122(45.0) | 60(53.1) | | 0.192 | 0.723(0.445–1.176) |
| | TT | 45(16.6) | 16(14.2) | | 0.999 | 1.001(0.506–1.981) |
| | CT+TT | 167(61.6) | 76(67.3) | 0.297(1.089) | 0.297 | 0.782(0.492–1.242) |
| rs2075800 | CC | 112(41.3) | 52(46.0) | 0.466(1.526) | Referent | 1.00 |
| | CT | 127(46.9) | 52(46.0) | | 0.547 | 1.134(0.715–1.798) |
| | TT | 32(11.8) | 9(8.0) | | 0.293 | 1.651(0.735–3.708) |
| | CT+TT | 159(58.7) | 61(54.0) | 0.397(0.717) | 0.398 | 1.210(0.778–1.883) |
| rs1380154 | CC | 78(28.8) | 30(26.5) | 0.905(0.199) | Referent | 1.00 |
| | CT | 147(54.2) | 63(55.8) | | 0.680 | 0.897(0.537–1.501) |
| | TT | 46(17.0) | 20(17.7) | | 0.721 | 0.885(0.451–1.734) |
| | CT+TT | 193(71.2) | 83(73.5) | 0.657(0.197) | 0.657 | 0.894(0.546–1.465) |
| rs391957 | CC | 121(44.6) | 57(50.4) | 0.451(1.593) | Referent | 1.00 |
| | CT | 128(47.2) | 50(44.2) | | 0.419 | 1.206(0.766–1.899) |
| | TT | 22(8.1) | 6(5.3) | | 0.263 | 1.727(0.664–4.493) |
| | CT+TT | 150(55.4) | 56(49.6) | 0.300(1.076) | 0.300 | 1.262(0.813–1.959) |
| rs17840761 | AA | 57(21.0) | 22(19.5) | 0.942(0.12) | Referent | 1.00 |
| | AG | 155(57.2) | 66(58.4) | | 0.736 | 0.906(0.513–1.603) |
| | GG | 59(21.8) | 25(22.1) | | 0.788 | 0.911(0.462–1.796) |
| | AG+GG | 214(79.0) | 91(80.5) | 0.730(0.119) | 0.730 | 0.908(0.524–1.573) |
| rs11218941 | AA | 75(27.7) | 41(36.3) | 0.060(5.64) | Referent | 1.00 |
| | AG | 134(49.4) | 57(50.4) | | 0.317 | 1.285(0.787–2.100) |
| | GG | 62(22.9) | 15(13.3) | | 0.019[*] | 2.260(1.144–4.462) |
| | AG+GG | 196(72.3) | 72(63.7) | 0.094(2.803) | 0.095 | 1.488(0.933–2.373) |
| rs10892958 | GG | 62(22.9) | 14(12.4) | 0.027(7.228)[*] | Referent | 1.00 |
| | GC | 134(49.4) | 56(49.6) | | 0.067 | 0.540(0.280–1.044) |
| | CC | 75(27.7) | 43(38.1) | | 0.008[*] | 0.394(0.197–0.786) |
| | GC+CC | 209(77.1) | 99(87.6) | 0.019(5.527)[*] | 0.021[*] | 0.477(0.255–0.893) |
| rs2236658 | CC | 45(16.6) | 9(8.0) | 0.053(5.867) | Referent | 1.00 |
| | CT | 130(48.0) | 54(47.8) | | 0.067 | 0.481(0.220–1.053) |
| | TT | 96(35.4) | 50(44.2) | | 0.018[*] | 0.384(0.174–0.849) |
| | CT+TT | 226(83.4) | 104(92.0) | 0.026(4.927)[*] | 0.030[*] | 0.435(0.205–0.922) |
| rs4802 | AA | 83(30.6) | 45(39.8) | 0.201(3.211) | Referent | 1.00 |
| | AG | 91(33.6) | 35(31.0) | | 0.206 | 1.410(0.828–2.401) |
| | GG | 97(35.8) | 33(29.2) | | 0.089 | 1.594(0.932–2.725) |
| | AG+GG | 188(69.4) | 68(60.2) | 0.082(3.035) | 0.082 | 1.499(0.949–2.367) |
| rs4936770 | TT | 98(36.2) | 33(29.2) | 0.261(2.69) | Referent | 1.00 |
| | TC | 129(47.6) | 55(48.7) | | 0.360 | 0.790(0.477–1.309) |
| | CC | 44(16.2) | 25(22.1) | | 0.103 | 0.593(0.316–1.112) |
| | TC+CC | 173(63.8) | 80(70.8) | 0.190(1.718) | 0.191 | 0.728(0.453–1.171) |
| rs1042665 | CC | 164(60.5) | 71(62.8) | 0.417(1.748) | Referent | 1.00 |

**Table 3** (*continued*)

| SNP | Genotype | Case, (%) | Control, (%) | *P* value[a] ($\chi^2$) | *P* value[b] | OR (95% CI) |
|---|---|---|---|---|---|---|
| | CT | 93(34.3) | 33(29.2) | | 0.422 | 1.220(0.751–1.982) |
| | TT | 14(5.2) | 9(8.0) | | 0.380 | 0.673(0.279–1.628) |
| | CT+TT | 107(39.5) | 42(37.2) | 0.671(0.180) | 0.671 | 1.103(0.701–1.734) |
| rs10873531 | AA | 165(60.9) | 66(58.4) | 0.074(5.22) | Referent | 1.00 |
| | AG | 97(35.8) | 37(32.7) | | 0.844 | 1.049(0.653–1.685) |
| | GG | 9(3.3) | 10(8.8) | | 0.034 | 0.360(0.140–0.926) |
| | AG+GG | 106(39.1) | 47(41.6) | 0.651(0.204) | 0.651 | 0.902(0.577–1.410) |
| rs8005905 | AA | 186(68.6) | 77(68.1) | 0.306(2.368) | Referent | 1.00 |
| | AT | 77(28.4) | 29(25.7) | | 0.712 | 1.099(0.665–1.817) |
| | TT | 8(3.0) | 7(6.2) | | 0.162 | 0.473(0.166–1.350) |
| | AT+TT | 85(31.4) | 36(31.9) | 0.924(0.009) | 0.924 | 0.977(0.610–1.566) |
| rs1165678 | AA | 191(70.5) | 72(63.7) | 0.378(1.945) | Referent | 1.00 |
| | AG | 72(26.6) | 38(33.6) | | 0.167 | 0.714(0.443–1.151) |
| | GG | 8(3.0) | 3(2.7) | | 0.994 | 1.005(0.259–3.894) |
| | AG+GG | 80(29.5) | 41(36.3) | 0.194(1.690) | 0.194 | 0.736(0.463–1.170) |
| rs1165681 | CC | 77(28.4) | 26(23.0) | 0.491(1.424) | Referent | 1.00 |
| | CT | 135(49.8) | 63(55.8) | | 0.237 | 0.724(0.423–1.236) |
| | TT | 59(21.8) | 24(21.2) | | 0.575 | 0.830(0.433–1.590) |
| | CT+TT | 194(71.6) | 87(77.0) | 0.276(1.187) | 0.277 | 0.753(0.451–1.256) |
| rs1882019 | GG | 131(48.3) | 60(53.1) | 0.479(1.471) | Referent | 1.00 |
| | GA | 114(42.1) | 46(40.7) | | 0.588 | 1.135(0.717–1.796) |
| | AA | 26(9.6) | 7(6.2) | | 0.241 | 1.701(0.699–4.137) |
| | GA+AA | 140(51.7) | 53(46.9) | 0.395(0.722) | 0.396 | 1.210(0.779–1.878) |
| rs3794241 | CC | 166(61.3) | 67(59.3) | 0.758(0.555) | Referent | 1.00 |
| | CT | 93(34.3) | 39(34.5) | | 0.873 | 0.962(0.602–1.539) |
| | TT | 12(4.4) | 7(6.4) | | 0.459 | 0.692(0.261–1.833) |
| | CT+TT | 105(38.7) | 46(40.7) | 0.720(0.129) | 0.720 | 0.921(0.589–1.442) |

**Notes.**

Abbreviations: CI, confidence interval; OR, odds ratio.

[a]*P* value for allele frequencies in cases and controls using 2-sided $\chi^2$ test.

[b]*P* values adjusted by age and gender using logistic regression.

*$P < 0.05$

rs2075800 (C>T), rs1380154 (C>T), rs391957 (C>T), rs17840761 (A>G), rs4936770 (T>C), HSP90 (rs1042665 (C>T), rs10873531 (A>G), rs8005905(A>T), rs1165678(A>G), rs1165681(C>T), rs1882019 (G>A), and rs3794241 (C>T) polymorphisms.

For rs10892958, the CC (OR = 0.394, 95% CI [0.197–0.786], and $p = 0.008$) and GC+CC genotypes (OR = 0.477, 95% CI [0.255–0.893], and $p = 0.021$) are associated with significantly decreased risks of CAD when the GG genotype is used as the reference. The C allele occurs less frequently in the CAD group (52.4%) than in the control group (62.8%, $p = 0.021$), thereby indicating that it might be a protective factor against the development of CAD. Similarly, the TT and CT+TT genotypes and T allele frequency of rs2236658 are significantly lower in the CAD cases than in the controls ($P = 0.018$, 0.030, 0.023, respectively). The GG genotype and G allele of the rs11218941 polymorphism are associated with an increased risk of CAD ($P = 0.019$ and $P = 0.021$, respectively). No

**Table 4 Allelic distribution of 17 HSP gene between all CAD patients (2n = 542) and Controls (2n = 226).**

| SNP | Allele | Case, (%) | Control, (%) | P value[a] ($\chi^2$) | P value[b] ($\chi^2$) | OR (95% CI) |
|---|---|---|---|---|---|---|
| rs2763979 | C | 330(60.9) | 134(59.3) | 0.681(0.169) | Referent | 1.00 |
| | T | 212(39.1) | 92(40.7) | | 0.681 | 0.936(0.682–1.284) |
| rs2075800 | C | 351(64.7) | 156(69.0) | 0.255(1.294) | Referent | 1.00 |
| | T | 191(35.3) | 70(31.0) | | 0.256 | 1.213(0.870–1.691) |
| rs1380154 | C | 303(55.9) | 123(54.4) | 0.707(0.141) | Referent | 1.00 |
| | T | 239(44.1) | 103(45.6) | | 0.707 | 0.942(0.690–1.287) |
| rs391957 | C | 370(68.3) | 164(72.6) | 0.238(1.392) | Referent | 1.00 |
| | T | 172(31.7) | 62(27.4) | | 0.238 | 1.230(0.872–1.734) |
| rs17840761 | A | 269(49.6) | 110(48.7) | 0.809(0.059) | Referent | 1.00 |
| | G | 273(50.4) | 116(51.3) | | 0.809 | 0.962(0.706–1.313) |
| rs11218941 | A | 284(52.4) | 139(61.5) | 0.021(5.345)[*] | Referent | 1.00 |
| | G | 258(47.6) | 87(38.5) | | 0.021[*] | 1.451(1.058–1.992) |
| rs10892958 | G | 258(47.6) | 84(37.2) | 0.008(7.029)[*] | Referent | 1.00 |
| | C | 284(52.4) | 142(62.8) | | 0.008[*] | 0.651(0.474–0.895) |
| rs2236658 | C | 220(40.6) | 72(31.9) | 0.023(5.161)[*] | Referent | 1.00 |
| | T | 322(59.4) | 154(68.1) | | 0.023[*] | 0.684(0.493–0.950) |
| rs4802 | A | 257(47.4) | 125(55.3) | 0.046(3.974)[*] | Referent | 1.00 |
| | G | 285(52.6) | 101(44.7) | | 0.047[*] | 1.372(1.005–1.875) |
| rs4936770 | T | 325(60.0) | 121(53.5) | 0.100(2.703) | Referent | 1.00 |
| | C | 217(40.0) | 105(46.5) | | 0.101 | 0.769(0.563–1.052) |
| rs1042665 | C | 421(77.7) | 175(77.4) | 0.942(0.005) | Referent | 1.00 |
| | T | 121(22.3) | 51(22.6) | | 0.942 | 0.986(0.680–1.430) |
| rs10873531 | A | 427(78.8) | 169(74.8) | 0.225(1.471) | Referent | 1.00 |
| | G | 115(21.2) | 57(25.2) | | 0.226 | 0.799(0.555–1.149) |
| rs8005905 | A | 449(82.8) | 183(81.0) | 0.537(0.382) | Referent | 1.00 |
| | T | 93(17.2) | 43(19.0) | | 0.537 | 0.881(0.591–1.315) |
| rs1165678 | A | 454(83.8) | 182(80.5) | 0.279(1.171) | Referent | 1.00 |
| | G | 88(16.2) | 44(19.5) | | 0.280 | 0.802(0.537–1.197) |
| rs1165681 | C | 289(53.3) | 115(50.9) | 0.538(0.380) | Referent | 1.00 |
| | T | 253(46.7) | 111(49.1) | | 0.538 | 0.907(0.665–1.237) |
| rs1882019 | G | 376(69.4) | 166(73.5) | 0.258(1.278) | Referent | 1.00 |
| | A | 166(30.6) | 60(26.5) | | 0.259 | 1.221(0.863–1.728) |
| rs3794241 | C | 425(78.4) | 173(76.5) | 0.571(0.322) | Referent | 1.00 |
| | T | 117(21.6) | 53(23.5) | | 0.571 | 1.359(0.954–1.937) |

**Notes.**

Abbreviations: CI, confidence interval; OR, odds ratio.

[a] P value for allele frequencies in cases and controls using 2-sided $\chi^2$ test.

[b] P values adjusted by age and gender using logistic regression.

[*] $P < 0.05$

statistically significant difference is observed in terms of the rs4802 genotype frequency, whereas a statistically significant difference is found between the G and A allele frequencies (OR = 1.372, 95% CI [1.005–1.875], and $p = 0.047$).
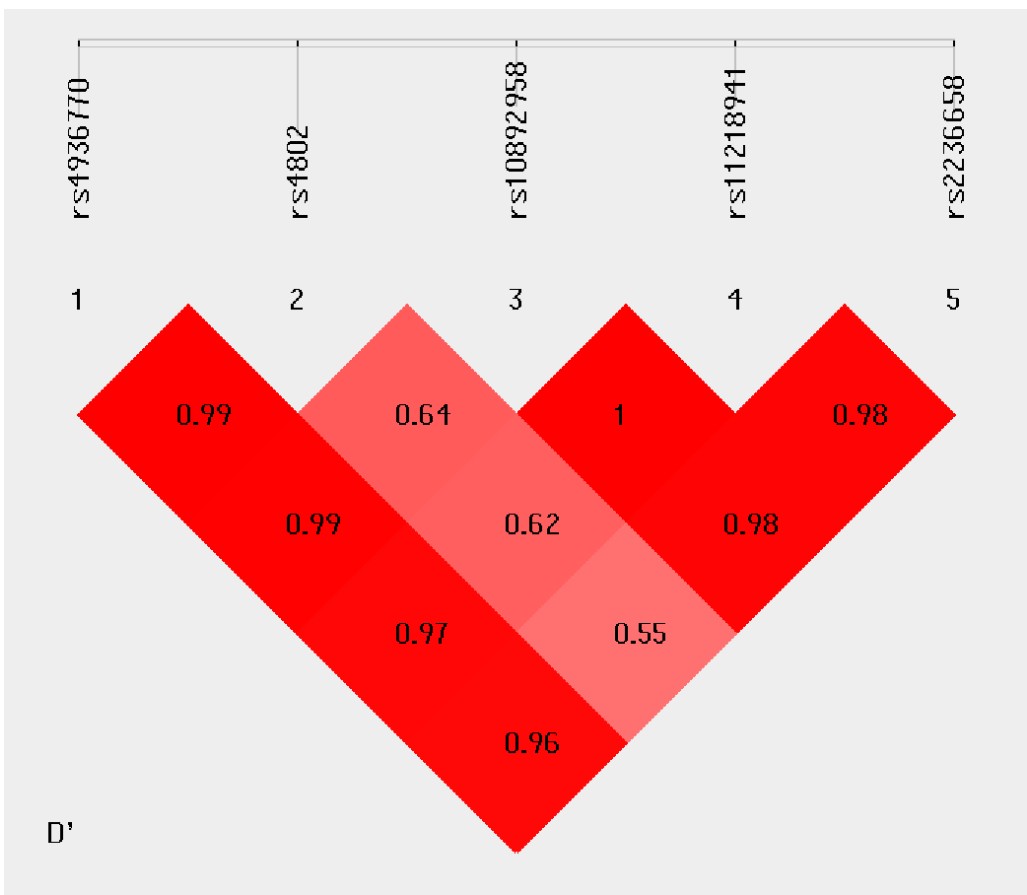

**Figure 1** Linkage disequilibrium pattern between five SNPs, rs4936770, rs4802, rs10892958, rs11218941 and rs2236658, in CAD patients and healthy controls.

## Haplotype Analysis

Haplotype analysis was conducted to investigate the linkage among the studied SNPs. Figure 1 shows that in the LD block of the HSPA8 gene, which codes the HSP70 family, rs4936770/ rs4802/ rs10892958/ rs11218941/ rs2236658 demonstrate a strong LD (rs4936770/rs4802: D′ = 0.99, $r^2$ = 0.72; rs4936770/ rs10892958: D′ = 0.99, $r^2$ = 0.57; rs4936770/ rs11218941: D′ = 0.97, $r^2$ = 0.55; rs4936770/ rs2236658: D′ = 0.96, $r^2$ = 0.41; rs10892958/ rs11218941: D′ = 1, $r^2$ = 0.98; rs10892958/ rs2236658: D′ = 0.98, $r^2$ = 0.73; rs11218941/ rs2236658: D′ = 0.98, $r^2$ = 0.72). The haplotype frequencies are presented in Table 5. The results show that haplotype TGGGC significantly increases the risk of CAD ($P$ = 0.008, OR = 1.609, 95% CI = 1.124–2.302).

## Influence of gene polymorphisms on the risk of depression/anxiety in CAD patients

The results of the association analysis between the studied polymorphisms and the risk of depression and anxiety among CAD patients are summarized in Tables 6 and 7, respectively.

**Table 5  Haplotype frequencies for *HSP70* polymorphisms in CAD and control group.**

| Haplotype (rs4936770/rs4802/ rs10892958/rs11218941/rs2236658) | CAD 2n = 542, (%) | Controls 2n = 226 (%) | OR (95% CI) | *P* value |
|---|---|---|---|---|
| TGGGC | 176(32.4) | 52(23.0) | 1.609 [1.124~2.302] | 0.008* |
| CACAT | 213(39.2) | 102(45.1) | 0.787 [0.575~1.076] | 0.134 |
| TGCAT | 68(12.5) | 37(16.3) | 0.732 [0.474~1.131] | 0.159 |
| TAGGC | 40(7.3) | 20(8.8) | 0.82 [0.468~1.437] | 0.489 |
| TGGGT | 41(7.5) | 12(5.3) | 1.459 [0.752~2.831] | 0.261 |

Notes.

CAD, coronary artery disease; CI, confidence interval; OR, odds ratio.

Haplotypes were omitted if the estimated haplotype frequency was < 3%.

*$P < 0.05$

No significant relationships are found among the CAD with depression (CAD+D) and CAD without depression (CAD −D) groups.

Among the CAD patients sub-grouped by comorbid anxiety, the presence of the AG (OR = 3.378, 95% CI [1.258–9.072], and $p = 0.016$) and GG genotypes (OR = 3.236, 95% CI [1.081–9.684], $p = 0.036$) of rs17840761 is associated with the increased risk of developing CAD comorbid anxiety compared with those without anxiety. Among the patients with CAD, the carriers of the CT genotype of rs1042665 have lower risk of anxiety than the carriers of the CC genotype (OR = 0.355, 95% CI [0.173–0.726], and $p = 0.005$), whereas the carriers of the CT genotype of rs1165681 demonstrate higher risk of anxiety than the carriers of the CC genotype (OR = 2.530, 95% CI [1.178–5.434], and $p = 0.017$). The risk of anxiety is significantly lower among CAD patients carrying the CT+TT genotypes of rs1042665 (OR = 0.334, 95% CI [0.167–0.667], and $p = 0.002$) and markedly higher (OR = 2.142, 95% CI [1.021–4.495], and $p = 0.044$) than those carrying the CC genotype of rs1165681.

We further examine the association of rs17840761, rs1042665, and rs1165681 with the severity of anxiety symptoms. The results are illustrated in Fig. 2. No significant difference is observed among the different genotypes of rs17840761. In addition, the GAD-7 scores of patients with the CC genotype of rs1042665 are markedly higher those of patients with the CT genotypes ($P < 0.016$). The rs1165681 polymorphism is also associated with the severity of anxiety symptoms due to the lower GAD-7 scores of CC genotype carriers than CT and TT genotype carriers ($p = 0.001$).

# DISCUSSION

Genetics study is an effective tool for understanding the underlying mechanism of CAD comorbid depression and anxiety. The present study investigates the relationship between HSP70/HSP90 polymorphisms and CAD comorbid depression and anxiety in a Chinese population.

We investigated the association of 11 HSP70 SNP polymorphisms with CAD susceptibility, and the results revealed that the rs4802, rs10892958, rs11218941, and rs2236658 gene variants are associated with CAD in the studied population. Further analysis suggested that these SNPs belonged to a component of HSPA8, which is an

Table 6 Genotypic distribution of 17 HSP polymorphisms among the CAD with depression group ($n = 123$) and CAD without depression group ($n = 148$).

| SNP | Genotype | CAD+D, (%) | CAD-D, (%) | P value[a] ($\chi^2$) | P value[b] | OR (95% CI) |
|---|---|---|---|---|---|---|
| rs2763979 | CC | 53(43.1) | 51(34.5) | 0.347(2.118) | Referent | |
| | CT | 51(41.5) | 71(48.0) | | 0.169 | 0.691(0.408–1.170) |
| | TT | 19(15.4) | 26(17.6) | | 0.328 | 0.703(0.347–1.424) |
| | CT+TT | 70(56.9) | 97(65.5) | 0.146(2.115) | 0.146 | 0.694(0.424–1.136) |
| rs2075800 | CC | 54(43.9) | 58(39.2) | 0.093(4.742) | Referent | 1.00 |
| | CT | 50(40.7) | 77(52.0) | | 0.169 | 0.697(0.417–1.166) |
| | TT | 19(15.4) | 13(8.8) | | 0.267 | 1.570(0.708–3.483) |
| | CT+TT | 69(56.1) | 90(60.8) | 0.433(0.615) | 0.433 | 0.823(0.507–1.338) |
| rs1380154 | CC | 41(33.3) | 37(25.0) | 0.102(4.562) | Referent | 1.00 |
| | CT | 58(47.2) | 89(60.1) | | 0.060 | 0.588(0.338–1.023) |
| | TT | 24(19.5) | 22(14.9) | | 0.966 | 0.984(0.475–2.042) |
| | CT+TT | 82(66.7) | 111(75.0) | 0.131(2.276) | 0.132 | 0.667(0.393–1.131) |
| rs391957 | CC | 49(39.8) | 72(48.6) | 0.241(2.848) | Referent | 1.00 |
| | CT | 65(52.8) | 63(42.6) | | 0.104 | 1.516(0.918–2.504) |
| | TT | 9(7.3) | 13(8.8) | | 0.971 | 1.017(0.404–2.563) |
| | CT+TT | 74(60.2) | 76(51.4) | 0.146(2.110) | 0.147 | 1.431(0.882–2.321) |
| rs17840761 | AA | 21(17.1) | 36(24.3) | 0.333(2.199) | Referent | 1.00 |
| | AG | 73(59.3) | 82(55.4) | | 0.184 | 1.526(0.818–2.848) |
| | GG | 29(23.6) | 30(20.3) | | 0.182 | 1.657(0.789–3.479) |
| | AG+GG | 102(82.9) | 112(75.7) | 0.145(2.126) | 0.147 | 1.561(0.856–2.849) |
| rs11218941 | AA | 36(29.3) | 39(26.4) | 0.328(2.230) | Referent | 1.00 |
| | AG | 64(52.0) | 70(47.3) | | 0.974 | 0.990(0.563–1.744) |
| | GG | 23(18.7) | 39(26.4) | | 0.201 | 0.639(0.322–1.269) |
| | AG+GG | 87(70.7) | 109(73.6) | 0.593(0.286) | 0.593 | 0.865(0.507–1.474) |
| rs10892958 | GG | 23(18.7) | 39(26.4) | 0.328(2.230) | Referent | 1.00 |
| | GC | 64(52.0) | 70(47.3) | | 0.164 | 1.550(0.837–2.873) |
| | CC | 36(29.3) | 39(26.4) | | 0.201 | 1.565(0.788–3.108) |
| | GC+CC | 100(81.3) | 109(73.6) | 0.135(2.229) | 0.137 | 1.556(0.869–2.785) |
| rs2236658 | CC | 17(13.8) | 28(18.9) | 0.381(1.930) | Referent | 1.00 |
| | CT | 64(52.0) | 66(44.6) | | 0.186 | 1.597(0.798–3.196) |
| | TT | 42(34.1) | 54(36.5) | | 0.503 | 1.281(0.620–2.645) |
| | CT+TT | 106(86.2) | 120(81.1) | 0.262(1.261) | 0.263 | 1.455(0.754–2.806) |
| rs4802 | AA | 43(35.0) | 40(27.0) | 0.326(2.245) | Referent | 1.00 |
| | AG | 37(30.1) | 54(36.5) | | 0.141 | 0.637(0.350–1.162) |
| | GG | 43(35.0) | 54(36.5) | | 0.317 | 0.741(0.411–1.334) |
| | AG+GG | 80(65.0) | 108(73.0) | 0.158(1.989) | 0.159 | 0.689(0.410–1.157) |
| rs4936770 | TT | 43(35.0) | 55(37.2) | 0.410(1.782) | Referent | 1.00 |
| | TC | 56(45.5) | 73(49.3) | | 0.944 | 0.981(0.578–1.666) |
| | CC | 24(19.5) | 20(13.5) | | 0.240 | 1.535(0.751–3.138) |
| | TC+CC | 80(65.0) | 93(62.8) | 0.707(0.141) | 0.707 | 1.100(0.668–1.811) |
| rs1042665 | CC | 80(65.0) | 84(56.8) | 0.246(2.806) | Referent | 1.00 |

(continued on next page)

**Table 6** (*continued*)

| SNP | Genotype | CAD+D, (%) | CAD-D, (%) | P value[a] (χ²) | P value[b] | OR (95% CI) |
|-----|----------|------------|------------|------------------|------------|-------------|
| | CT | 39(31.7) | 54(36.5) | | 0.291 | 0.758(0.454–1.267) |
| | TT | 4(3.3) | 10(6.8) | | 0.156 | 0.420(0.127–1.393) |
| | CT+TT | 43(35.0) | 64(43.2) | 0.165(1.929) | 0.165 | 0.705(0.431–1.155) |
| rs10873531 | AA | 74(60.2) | 91(61.5) | 0.702(0.709) | Referent | 1.00 |
| | AG | 46(37.4) | 51(34.5) | | 0.686 | 1.109(0.671–1.834) |
| | GG | 3(2.4) | 6(4.1) | | 0.502 | 0.615(0.149–2.542) |
| | AG+GG | 49(39.8) | 57(38.5) | 0.824(0.049) | 0.824 | 1.057(0.648–1.725) |
| rs8005905 | AA | 84(68.3) | 102(68.9) | 0.877(0.263) | Referent | 1.00 |
| | AT | 36(29.3) | 41(27.7) | | 0.814 | 1.066(0.626–1.816) |
| | TT | 3(2.4) | 5(3.4) | | 0.671 | 0.729(0.169–3.138) |
| | AT+TT | 39(31.7) | 46(31.1) | 0.912(0.012) | 0.912 | 1.030(0.615–1.723) |
| rs1165678 | AA | 83(67.5) | 108(73.0) | 0.614(0.974) | Referent | 1.00 |
| | AG | 36(29.3) | 36(24.3) | | 0.342 | 1.301(0.756–2.240) |
| | GG | 4(3.3) | 4(2.7) | | 0.715 | 1.301(0.316–5.357) |
| | AG+GG | 40(32.5) | 40(27.0) | 0.324(0.974) | 0.324 | 1.301(0.771–2.196) |
| rs1165681 | CC | 31(25.2) | 46(31.1) | 0.466(1.526) | Referent | 1.00 |
| | CT | 66(53.7) | 69(46.6) | | 0.226 | 1.419(0.805–2.502) |
| | TT | 26(21.1) | 33(22.3) | | 0.656 | 1.169(0.588–2.323) |
| | CT+TT | 92(74.8) | 102(68.9) | 0.285(1.141) | 0.286 | 1.338(0.783–2.287) |
| rs1882019 | GG | 60(48.8) | 71(48.0) | 0.753(0.568) | Referent | 1.00 |
| | GA | 53(43.1) | 61(41.2) | | 0.914 | 1.028(0.621–1.701) |
| | AA | 10(8.1) | 16(10.8) | | 0.493 | 0.740(0.312–1.751) |
| | GA+AA | 63(51.2) | 77(52.0) | 0.895(0.018) | 0.895 | 0.968(0.600–1.562) |
| rs3794241 | CC | 78(63.4) | 88(59.5) | 0.697(0.722) | Referent | 1.00 |
| | CT | 39(31.7) | 54(36.5) | | 0.433 | 0.815(0.488–1.360) |
| | TT | 6(4.9) | 6(4.1) | | 0.840 | 1.128(0.349–3.642) |
| | CT+TT | 45(36.6) | 60(40.5) | 0.506(0.443) | 0.506 | 0.846(0.517–1.384) |

**Notes.**
CAD+D, CAD with depression; CAD-D, CAD without depression.
[a]P value for genotype frequencies in CAD+D and CAD-D using 2-sided $\chi^2$ test.
[b]P values adjusted by smoking and drinking habit, hypertension, diabetes mellitus, insomnia and stroke history using binary logistic regression.

important member of the HSP70 family. HSPA8 is also referred to as HSC70, HSC71, HSP71, or HSP73 and represents a constitutively expressed cognate protein of the HSP70 family. HSPA8 is an essential protein in the housekeeping of the HSP70 family and plays a major role in the protein quality control, with its chaperone in the folding protein process. *He et al. (2010)* identified four tag SNPs (rs2236659, rs2276077, rs10892958, and rs1461496) of HSPA8 and evaluated their association with the susceptibility to CHD. The findings showed that the promoter SNP rs2236659 was associated with the susceptibility to CHD, and the carriers of the C allele of rs2236659 possessed a decreased CHD risk. By contrast, rs10892958 did not exhibit any association with the susceptibility to CHD, which contradicted our conclusion that the TT and CT+TT genotypes and T allele frequency of rs2236658 were significantly lower in the CAD cases than in the controls.

Further haplotype analysis showed that the HSPA8 SNPs in this study displayed strong LD, whereas the other SNPs did not. These results were also inconsistent with the findings
**Table 7** Genotypic distribution of 17 HSP polymorphisms among the CAD with anxiety group ($n = 57$) and CAD without anxiety group ($n = 214$).

| SNP | Genotype | CAD+A,(%) | CAD-A,(%) | P value[a] ($\chi^2$) | P value[b] | OR (95% CI) |
|---|---|---|---|---|---|---|
| rs2763979 | CC | 25(43.9) | 79(36.9) | 0.190(3.321) | Referent | 1.00 |
| | CT | 27(47.4) | 95(44.4) | | 0.734 | 0.898(0.483–1.670) |
| | TT | 5(8.8) | 40(18.7) | | 0.078 | 0.395(0.141–1.109) |
| | CT+TT | 32(56.1) | 135(63.1) | 0.338(0.918) | 0.339 | 0.749(0.414–1.354) |
| rs2075800 | CC | 20(35.1) | 92(43.0) | 0.438(1.652) | Referent | 1.00 |
| | CT | 31(54.4) | 96(44.9) | | 0.219 | 1.485(0.791–2.791) |
| | TT | 6(10.5) | 26(12.1) | | 0.908 | 1.062(0.386–2.917) |
| | CT+TT | 37(64.9) | 122(57.0) | 0.282(1.159) | 0.283 | 1.395(0.760–2.561) |
| rs1380154 | CC | 17(29.8) | 61(28.5) | 0.963(0.076) | Referent | 1.00 |
| | CT | 30(52.6) | 117(54.7) | | 0.808 | 0.920(0.470–1.799) |
| | TT | 10(17.5) | 36(16.8) | | 0.994 | 0.997(0.412–2.410) |
| | CT+TT | 40(70.2) | 153(71.5) | 0.845(0.038) | 0.845 | 0.938(0.494–1.780) |
| rs391957 | CC | 26(45.6) | 95(44.4) | 0.955(0.093) | Referent | 1.00 |
| | CT | 26(45.6) | 102(47.7) | | 0.820 | 0.931(0.505–1.716) |
| | TT | 5(8.8) | 17(7.9) | | 0.897 | 1.075(0.362–3.188) |
| | CT+TT | 31(54.4) | 119(55.6) | 0.869(0.027) | 0.869 | 0.952(0.529–1.712) |
| rs17840761 | AA | 5(8.8) | 52(24.3) | 0.038(6.550)* | Referent | 1.00 |
| | AG | 38(66.7) | 117(54.7) | | 0.016* | 3.378(1.258–9.072) |
| | GG | 14(24.6) | 45(21.0) | | 0.036* | 3.236(1.081–9.684) |
| | AG+GG | 52(91.2) | 162(75.7) | 0.011(6.534)* | 0.015* | 3.338(1.266–8.801) |
| rs11218941 | AA | 15(26.3) | 60(28.0) | 0.449(1.600) | Referent | 1.00 |
| | AG | 32(56.1) | 102(47.7) | | 0.520 | 1.255(0.629–2.505) |
| | GG | 10(17.5) | 52(24.3) | | 0.560 | 0.769(0.318–1.858) |
| | AG+GG | 42(73.7) | 154(72.0) | 0.796(0.067) | 0.796 | 1.091(0.563–2.112) |
| rs10892958 | GG | 10(17.5) | 52(24.3) | 0.449(1.600) | Referent | 1.00 |
| | GC | 32(56.1) | 102(47.7) | | 0.222 | 1.631(0.744–3.575) |
| | CC | 15(26.3) | 60(28.0) | | 0.560 | 1.300(0.538–3.141) |
| | GC+CC | 47(82.5) | 162(75.7) | 0.281(1.164) | 0.283 | 1.509(0.712–3.196) |
| rs2236658 | CC | 8(14.0) | 37(17.3) | 0.704(0.703) | Referent | 1.00 |
| | CT | 30(52.6) | 100(46.7) | | 0.459 | 1.387(0.583–3.300) |
| | TT | 19(33.3) | 77(36.0) | | 0.777 | 1.141(0.457–2.848) |
| | CT+TT | 49(86.0) | 177(82.7) | 0.557(0.344) | 0.558 | 1.280(0.560–2.928) |
| rs4802 | AA | 16(28.1) | 67(31.3) | 0.296(2.435) | Referent | 1.00 |
| | AG | 24(42.1) | 67(31.3) | | 0.268 | 1.500(0.732–3.074) |
| | GG | 17(29.8) | 80(37.4) | | 0.762 | 0.890(0.418–1.895) |
| | AG+GG | 41(71.9) | 147(68.7) | 0.637(0.222) | 0.638 | 1.168(0.612–2.228) |
| rs4936770 | TT | 17(29.8) | 81(37.9) | 0.346(2.123) | Referent | 1.00 |
| | TC | 32(56.1) | 97(45.3) | | 0.178 | 1.572(0.814–3.035) |
| | CC | 8(14.0) | 36(16.8) | | 0.904 | 1.059(0.419–2.677) |
| | TC+CC | 40(70.2) | 133(62.1) | 0.262(1.256) | 0.264 | 1.433(0.762–2.694) |
| rs1042665 | CC | 45(78.9) | 119(55.6) | 0.005(10.423)* | Referent | 1.00 |

*(continued on next page)*

**Table 7** (*continued*)

| SNP | Genotype | CAD+A,(%) | CAD-A,(%) | *P* value[a] ($\chi^2$) | *P* value[b] | OR (95% CI) |
|---|---|---|---|---|---|---|
| | CT | 11(19.3) | 82(38.3) | | 0.005* | 0.355(0.173–0.726) |
| | TT | 1(1.8) | 13(6.1) | | 0.130 | 0.203(0.026–1.600) |
| | CT+TT | 12(21.1) | 95(44.4) | 0.001(10.262)* | 0.002* | 0.334(0.167–0.667) |
| rs10873531 | AA | 41(71.9) | 124(57.9) | 0.150(3.791) | Referent | 1.00 |
| | AG | 15(26.3) | 82(38.3) | | 0.076 | 0.553(0.288–1.064) |
| | GG | 1(1.8) | 8(3.7) | | 0.366 | 0.378(0.046–3.114) |
| | AG+GG | 16(28.1) | 90(42.1) | 0.055(3.697) | 0.057 | 0.538(0.284–1.018) |
| rs8005905 | AA | 42(73.7) | 144(67.3) | 0.608(0.996) | Referent | 1.00 |
| | AT | 14(24.6) | 63(29.4) | | 0.429 | 0.762(0.389–1.494) |
| | TT | 1(1.8) | 7(3.3) | | 0.510 | 0.490(0.059–4.094) |
| | AT+TT | 15(26.3) | 70(32.7) | 0.355(0.855) | 0.356 | 0.735(0.382–1.414) |
| rs1165678 | AA | 39(68.4) | 152(71.0) | 0.914(0.180) | Referent | 1.00 |
| | AG | 16(28.1) | 56(26.2) | | 0.749 | 1.114(0.577–2.149) |
| | GG | 2(3.5) | 6(2.8) | | 0.754 | 1.299(0.252–6.687) |
| | AG+GG | 18(31.6) | 62(29.0) | 0.701(0.147) | 0.701 | 1.132(0.602–2.128) |
| rs1165681 | CC | 10(17.5) | 67(31.3) | 0.032(6.896)* | Referent | 1.00 |
| | CT | 37(64.9) | 98(45.8) | | 0.017* | 2.530(1.178–5.434) |
| | TT | 10(17.5) | 49(22.9) | | 0.519 | 1.367(0.528–3.538) |
| | CT+TT | 47(82.5) | 147(68.7) | 0.041(4.193)* | 0.044* | 2.142(1.021–4.495) |
| rs1882019 | GG | 25(43.9) | 106(49.5) | 0.748(0.581) | Referent | 1.00 |
| | GA | 26(45.6) | 88(41.1) | | 0.474 | 1.253(0.676–2.323) |
| | AA | 6(10.5) | 20(9.3) | | 0.641 | 1.272(0.463–3.496) |
| | GA+AA | 32(56.1) | 108(50.5) | 0.446(0.580) | 0.447 | 1.256(0.698–2.261) |
| rs3794241 | CC | 37(64.9) | 129(60.3) | 0.797(0.453) | Referent | 1.00 |
| | CT | 18(31.6) | 75(35.0) | | 0.580 | 0.837(0.445–1.573) |
| | TT | 2(3.5) | 10(4.7) | | 0.651 | 0.697(0.146–3.324) |
| | CT+TT | 20(35.1) | 85(39.7) | 0.524(0.407) | 0.524 | 0.820(0.446–1.508) |

**Notes.**

CAD+A, CAD with anxiety; CAD-A, CAD without anxiety.

[a]*P* value for genotype frequencies in CAD+A and CAD-A using 2-sided $\chi^2$ test.

[b]*P* values adjusted by smoking and drinking habit, hypertension, diabetes mellitus, insomnia and stroke history using binary logistic regression.

*Significant difference (P < 0.05)

of a study on a Chinese population, which indicated that a haplotype of HSPA8 SNPs contributes to a lower CHD risk compared with the common haplotype (*He et al., 2010*). Studies showed that HSPA8 acted as an accessory protein of a hyperpolarization-activated chloride channel from rat pulmonary vein cardiomyocytes (*Okamoto & Nagasawa, 2019*). In addition, the HSPA8 chaperone system played a major role in regulating the cardiac myosin binding protein C (*Glazier et al., 2018*). This inference could partly explain the obtained results. A genome-wide association study reported that HSPA8 is associated with the new onset of atrial fibrillation, MI, acute kidney injury, and stroke after cardiac surgery, thereby indicating its potential role in cardiac protection (*Westphal et al., 2019*). Furthermore, He et al. suggested that HSPA8 might take part in the development of CHD in two ways. First, HSPA8 could protect against reactive oxygen species with other

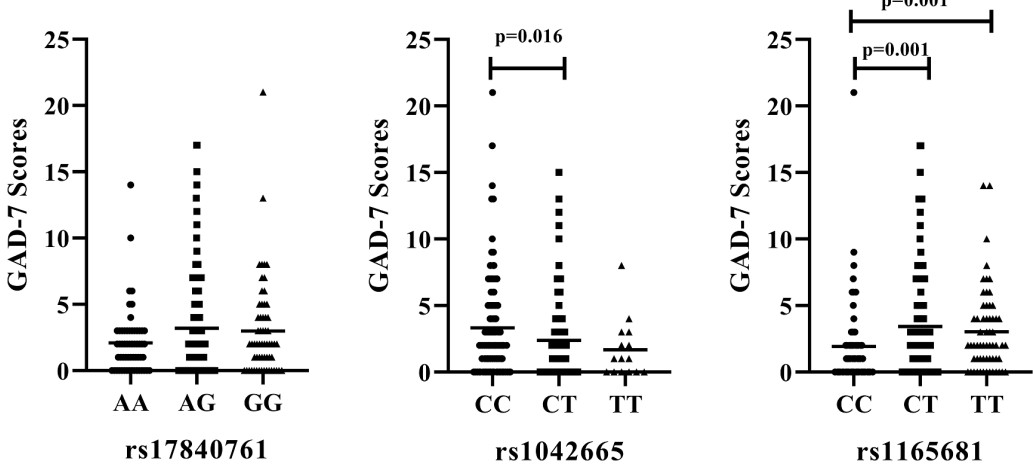

**Figure 2 Association of HSP polymorphisms and GAD-7 scores in CAD patients with comorbid anxiety.** (A) rs178040761 (B) rs1042665 and (C) rs1165681. GAD-7, Generalized Anxiety Disorder -7.

HSPs, which were involved in the etiology of CHD. Second, HSPA8 could protect against hypoxia-induced apoptosis in hypoxia-induced apoptosis-resistant macrophages.

HSP90 played a myocardial protective role in the crosslinking with actin filaments in a $Ca^{2+}$- and ATP-dependent manner, thereby helping the modulation of the cytoskeletal dynamics and its positive signal in the cytoplasm of myocardial cells after heat stress (*Islam et al., 2014*). Our previous study revealed a remarkable association between FKBP5 gene variations and CAD risk. The most perceived molecular function of FKBP5 was its role as a co-chaperone of HSP90 heterocomplexes. Therefore, we investigated the relationship between HSP90 polymorphisms and CAD. However, the results showed no significant difference in the HSP90 SNP between the CAD patients and healthy individuals in this study. The limited sample size and single species restricted the scope of the present study. Hence, further studies with larger samples from different races should be conducted to provide more evidence.

In this study, we focused on the influence of HSP70/HSP90 gene polymorphisms on the risk of depression/anxiety in CAD patients. Our data suggested that genetic polymorphisms of HSPs have no significant impact on comorbid depression. This result was consistent with a report on a Chinese population, which highlighted that no significant association exists between HSP70 gene (rs2075799) SNPs and the comorbidity of CAD with depression (*Min, Shunying & Xiaoyan, 2017*). However, the different genotypes of rs17840761, rs1042665, and rs1165681 presented anxiety risk among the CAD patients. The presence of the AG and GG genotypes of rs17840761, CT and CT+TT genotype of rs1165681 were associated with the increased risk of developing CAD comorbid with anxiety compared with those without anxiety. The carriers of the CT genotype of rs1042665 had lower risk of anxiety than the carriers of the CC genotype. These paradoxical results might be explained by the relatively limited sample size, which was the major limitation of the present study.

Moreover, the regional and racial biases, as well as the lack of correction for potential population stratification were also the potential limitations for this study.

## CONCLUSIONS

The results suggest the association between HSPA8 gene polymorphism and CAD in a Chinese population. The rs17840761 of HSP70 and rs1042665/rs1165681 of HSP90 gene polymorphisms may influence the comorbidity of anxiety among CAD patients. However, the limited sample size, especially the healthy subjects, may urge us to draw conservative conclusions for the conclusion for the association between HSP70/HSP90 gene polymorphisms and CAD risks. Therefore, further investigations through replication studies with large samples are necessary to confirm these conclusions and clarify the role of HSP polymorphisms in the pathogenesis of the comorbidity of anxiety/depression and CAD.

### Funding
This work was supported by the National Natural Science Foundation of China (grant number 81602846), the Chinese Society of Clinical Pharmacy Project of Wu Jieping Medical Foundation (grant number 320.6750.19090-8), the Taishan Scholar Project of Shandong Province (grant number tsqn201812159), The Key Research and Development Program of Jining Science and Technology (2019SMNS012) and the Scientific Research Development Fundation of Kangda College of Nanjing Medical University (KD2020KYJJYB044). The funders had no role in study design, data collection and analysis, decision to publish, or preparation of the manuscript.

### Grant Disclosures
The following grant information was disclosed by the authors:
National Natural Science Foundation of China: 81602846.
Chinese Society of Clinical Pharmacy Project of Wu Jieping Medical Foundation: 320.6750.19090-8.
Taishan Scholar Project of Shandong Province: tsqn201812159.
The Key Research and Development Program of Jining Science and Technology: 2019SMNS012.
Scientific Research Development Fundation of Kangda College of Nanjing Medical University: KD2020KYJJYB044.

### Competing Interests
The authors declare there are no competing interests.

### Author Contributions
- Haidong Wang analyzed the data, prepared figures and/or tables, authored or reviewed drafts of the paper, and approved the final draft.

- Yudong Ba analyzed the data, prepared figures and/or tables, and approved the final draft.
- Wenxiu Han and Haixia Zhang performed the experiments, authored or reviewed drafts of the paper, and approved the final draft.
- Laiqing Zhu conceived and designed the experiments, prepared figures and/or tables, and approved the final draft.
- Pei Jiang conceived and designed the experiments, performed the experiments, analyzed the data, prepared figures and/or tables, authored or reviewed drafts of the paper, and approved the final draft.

## Human Ethics

The following information was supplied relating to ethical approvals (i.e., approving body and any reference numbers):

This study was carried out in accordance with the recommendations of the medical ethics committee of Jining First People's Hospital guidelines with written informed consent from all subjects. All subjects gave written informed consents and the protocol was approved by the medical ethics committee of the Jining First People's Hospital (JY2016062).

## Data Availability

The raw data are available in the Supplemental File.

## Supplemental Information

Supplemental information for this article can be found online at http://dx.doi.org/10.7717/peerj.11636#supplemental-information.

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
