# Peer review of "Association of heat shock protein polymorphisms with patient susceptibility to coronary artery disease comorbid depression and anxiety in a Chinese population"

_PeerJ, doi:10.7717/peerj.11636_

## Round 0.1 · original submission · Major Revisions

Your manuscript was considered interesting and valuable, but one of the reviewers raised some important comments that need to be addressed. Specifically, the reviewer stated that a comparison between healthy controls and CAD patients should be included in your analysis. Additionally, a binary logistic regression analysis should be performed to identify variables independently associated with CAD, and your findings should be addressed in the discussion.

Please, submit a detailed rebuttal which shows where and how you have taken all comments and suggestions into consideration. If you do not agree with some of the reviewers’ comments or suggestions, please explain why. Your rebuttal will be critical in making a final decision on your manuscript. Please, note also that your revised version may enter a new round of review by the same or by different reviewers. Therefore, I cannot guarantee that your manuscript will eventually be accepted.

·

Basic reporting

The paper by H. Wang et al deals with the interesting topic of the gene polymorphisms of HSP70 and HSP90 and their association with comorbid anxiety and depression in Coronary Artery Disease (CAD).

Language:
Despite the good use of English language, there are some points that need linguistic revision :
ll.50-51: the verb is missing before "the higher risks of anxiety" as well as the ending "adjusted".
ll.106: the sentence begins with a reference in parentheses.
ll. 108: the sentence begins with the expression "via inhibiting.." which needs rephrasing.
ll.112: the sentence starts with the expression "in a group of patients.." without being clear to which group the sentence refers to.
ll.124: again a sentence begins with a reference in parentheses.
ll.130: misspelling of "anxiety"
ll.135-136: " overviewed the association" and "proposed that four ..."
ll.269 : "whether adjusted"
Table 2 - "insomnia"
Figure 2- I believe that GAD should read GAD-7 and not GAD-9 in the title.

Literature references:
Although well referenced there are some lines where there is mentioning of numerous studies with no references being presented :
ll.89-90; ll.103; ll.106; ll145.

Professional article structure

Well structured, however in the abstract in line 50, the reference to rs1042665 is read at first as belonging to HSP70 polymorphisms and not to HSP90 as is shortly afterwards explained.I believe it should be clarified when initially presenting this polymorphism that it belongs to HSP90.
Moreover, in the abstract the absence of correlation between comorbid depression and HSP gene polymorphism in CAD, leaves the research questions partially unanswered in the results section. I think it should be included in the abstract that no association between depression and gene polymorphism was found in this research.
The raw data, tables are really sound.I

Experimental design

It is an interesting paper that serves the aims and scopes of the journal.
The research question was well defined and covered an interesting topic of the multidimensional functions of heat shock proteins.
Ethical and technical high standards were followed.
In the methods question, was the diagnosis of depression and anxiety simultaneously completed by the two psychiatrists? Was the diagnosis only based on the two mentioned inventories? If separate interviews were conducted, was there an inter-scorer reliability being taken into consideration?

Validity of the findings

No comment

Reviewer 2 ·

Basic reporting

The present manuscript provides important information on Heat shock proteins (HSPs) and the association between HSP70 and HSP90 with CAD patients with depression or anxiety in a Chinese population. In particular, the results indicate that HSP/HSP90 gene polymorphism are associated with anxiety but not depression in CAD patients.

However, before the manuscript can be considered for publication, the authors should address some major points.

Experimental design

• Please provide explanation related to the criteria for the selection of specific SNPs.
• Please clarify the number of healthy participants included in the present study (113 derived from a priori statistics versus 171 healthy volunteers). Please explain why the number of healthy participants is lower as compared to CAD patients. In this type of association studies at least the same number of participants is preferred.
• Present findings present sample characteristics and appropriate comparisons, SNP association analyses (case versus control) but comparison between CAD patients and healthy participants concerning depression or anxiety state is missing. Subsequently a binary logistic regression analysis should be performed to access the variables independently associated with CAD based on the statistically significant results derived from previous comparisons between patients and healthy participants (independent variables for equation).

Validity of the findings

Taken into consideration the afore mentioned points the findings of the present study could be strengthen and discussion section will be essentially enriched.

Conclusions are well stated but limited since the main comparisons/findings concern subgroups within GAD patients .

Additional comments

The authors should thoroughly address the afore mentioned points.

---

## Round 0.2 · Minor Revisions

You addressed the reviewers' comments to their satisfaction, however one of the reviewers has a minor comment on restructuring the discussion of your manuscript.

Please, submit a detailed rebuttal which shows where and how you have taken the above comment and suggestions into consideration. If you do not agree with the reviewer's comment please explain why. Your rebuttal will be critical in making a final decision on your manuscript. Please, note also that your revised version may enter a new round of review by the same or by different reviewers. Therefore, I cannot guarantee that your manuscript will eventually be accepted.

·

Basic reporting

Discussion should be restructured:
Lines 302-312 and 352-363 better belong to the introduction as part of the literature review and the initial statement of the basic hypotheses . In the discussion section, these lines seem as reiteration of the introductory part of the paper, not promoting the conclusive nature of discussion.

Experimental design

no comment

Validity of the findings

no comment

Additional comments

Discussion needs restructuring in order to avoid unnecessary reiterations of literature review and statement of basic hypotheses.

Reviewer 2 ·

Basic reporting

no comment

Experimental design

no comment

Validity of the findings

no comment

Additional comments

Thea authors have addressed all comments raised by reviewer

---

## Round 0.3 · accepted · Accept

Thank you for incorporating all the reviewers' comments. As a result, your manuscript is much improved.

·

Basic reporting

no comment

Experimental design

no comment

Validity of the findings

no comment

Additional comments

You did very good work with the revisions.